# Interaction between Nanoparticles, Membranes and Proteins: A Surface Plasmon Resonance Study

**DOI:** 10.3390/ijms24010591

**Published:** 2022-12-29

**Authors:** Erenildo Ferreira de Macedo, Nivia Salles Santos, Lucca Silva Nascimento, Raphaël Mathey, Sophie Brenet, Matheus Sacilotto de Moura, Yanxia Hou, Dayane Batista Tada

**Affiliations:** 1Laboratory of Nanomaterials and Nanotoxicology, Instituto de Ciência e Tecnologia, Universidade Federal de São Paulo (UNIFESP), São José dos Campos 12231280, SP, Brazil; 2University Grenoble Alpes, CEA, CNRS, IRIG-SYMMES, 38000 Grenoble, France

**Keywords:** nanoparticles, protein corona, membranes, surface plasmon resonance imaging, biomolecular interactions, blood serum proteins, nanocomplex dissociation rate, affinity

## Abstract

Regardless of the promising use of nanoparticles (NPs) in biomedical applications, several toxic effects have increased the concerns about the safety of these nanomaterials. Although the pathways for NPs toxicity are diverse and dependent upon many parameters such as the nature of the nanoparticle and the biochemical environment, numerous studies have provided evidence that direct contact between NPs and biomolecules or cell membranes leads to cell inactivation or damage and may be a primary mechanism for cytotoxicity. In such a context, this work focused on developing a fast and accurate method to characterize the interaction between NPs, proteins and lipidic membranes by surface plasmon resonance imaging (SPRi) technique. The interaction of gold NPs with mimetic membranes was evaluated by monitoring the variation of reflectivity after several consecutive gold NPs injections on the lipidic membranes prepared on the SPRi biochip. The interaction on the membranes with varied lipidic composition was compared regarding the total surface concentration density of gold NPs adsorbed on them. Then, the interaction of gold and silver NPs with blood proteins was analyzed regarding their kinetic profile of the association/dissociation and dissociation constants (k_off_). The surface concentration density on the membrane composed of 1-palmitoyl-2-oleoyl-glycero-3-phosphocholine and cholesterol (POPC/cholesterol) was 2.5 times higher than the value found after the injections of gold NPs on POPC only or with dimethyldioctadecylammonium (POPC/DDAB). Regarding the proteins, gold NPs showed preferential binding to fibrinogen resulting in a value of the variation of reflectivity that was 8 times higher than the value found for the other proteins. Differently, silver NPs showed similar interaction on all the tested proteins but with a variation of reflectivity on immunoglobulin G (IgG) 2 times higher than the value found for the other tested proteins.

## 1. Introduction

Nanoparticles (NPs) of diverse sizes, shapes and compositions have been developed for increasing applications in materials and life sciences [1,2]. Particularly in cancer therapy, NPs physico-chemical properties allow them to target cancer cells, enhancing the penetration and accumulation of drugs in tumor tissues and reducing the systemic effects of usual therapies [3,4,5].

When NPs are used in biomedical applications, they are brought in direct contact with biological fluids, wherein they assume the biological identity that can bring new features not predicted during the development process of the NPs [4,6,7]. It has been continuously proven that once NPs assume their biological identity, they can provide unexpected therapeutic efficiency or toxicity. For this reason, a full characterization of the interaction between NPs and the biological components is crucial to evaluate the NPs therapeutic efficiency as well as their potential toxicity [6,8,9].

In the last years, several research groups have focused their works on understanding the physicochemical changes that NPs go through in the biological medium [5,6] and how they interact with membranes and cells. The adsorption of NPs on the cell membrane or microbial cell wall and how they affect the integrity of these membranes have been shown to depend on the cell type, NPs size, NPs surface properties and NPs colloidal state [10,11,12,13]. For example, gold NPs smaller than 6 nm were shown to enter the nucleus, whereas larger NPs could only be found in the cytoplasm, which could explain the increased cytotoxicity of ultrasmall gold NPs [14]. The interaction between cells and NPs has also been used to promote cell differentiation on the surface of scaffolds or implants, improving bioactivity and tissue regeneration [15,16]. In the bloodstream, there is a high content and wide diversity of proteins, and immediately after the contact of NPs with the bloodstream, a coating of these proteins is formed surrounding the surface of the NPs. As a result, a protein corona is formed on the NPs surface, whose composition depends on the NPs size, surface chemistry and medium conditions [4,6,7,13,17,18]. Much evidence has been reported on the effects of protein corona on cell targeting, intracellular fate and cell uptake process [6,7,13,19]. Protein corona can affect the NP’s hydrophobic character or even activate an internalization process triggered by a specific bio-interaction [20,21,22,23,24].

Due to the aforementioned reasons, the evaluation of the interaction of NPs with membranes and proteins should be a mandatory step in the development process of an NP. This investigation is crucial to predicting and understanding the NPs’ behavior in the biological medium. Therefore, it can help guarantee their performance in vivo. The analysis of the biological interaction of NPs also finds application in the field of materials science since it can allow associating NPs’ physicochemical properties, the formation of protein corona and the NP effects on membranes. Through this type of analysis, it could be possible to identify which NP properties should be controlled in order to tune NPs’ behavior in the biological medium.

It is clear that the NPs surface properties mediate protein corona formation and NPs interaction with membranes. Therefore, it is an important aspect to be considered in the design of NPs. Different surface coating of NP can be engineered to modulate the interaction with cell membrane, possibly targeting a specific membrane microdomain and even driving the NPs through a specific mechanism of cell uptake. Innovative NPs coating could also control the protein corona formation and composition. Gorshkov et al. [25] reported that the binding of proteins on the surface of silver NPs would be observed after an unfolding of a specific site in the protein. The main components of protein corona of silver NPs were identified to be proteins with a tendency to prefer a higher amount of β-sheet formation and hydrophobicity. The influence of surface chemistry of NPs on the composition of protein corona was also evidenced by several groups that also pointed out the effects of medium conditions. Eigenheer et al. showed that at low ion concentrations, electrostatic interactions are the driving forces that define protein corona formation [26]. In other work, the composition of the protein corona was shown to change as a result of different binding affinity of the proteins available in the medium, which could provide a strategy to change protein corona composition by host-guest interaction [6,27,28].

NP interaction with membranes has been extensively characterized by computational simulation. Additionally, experimental techniques have been used to complement the computational assays, such as atomic force microscopy (AFM), differential scanning calorimetry and giant vesicles coupled with optical microscopy. Nevertheless, current computational tools do not allow modeling the complexity of the nanoparticle-cell membrane interaction. For this reason, the experimental approaches must be designed as simply as possible to be modeled [10,29,30,31,32].

The composition of the protein corona of gold NPs and silver NPs in contact with blood serum^8^ or plasma has already been characterized by mass spectrometry, liquid chromatography and SDS-PAGE. Most reported experimental methodologies were based on a combination of different techniques to indirectly identify the composition of the protein corona. These approaches detected the proteins that were left in the medium after the incubation with NPs followed by a centrifugation step. The proteins were also identified after being washed from the precipitated NPs. Despite extensive information, these methods are too complex and time-consuming to be considered a routine step in the development process of NPs. Besides, as some of the researchers adverted, the accuracy of their approaches could be compromised by the limit of detection. Gorshkov et al., for example, mentioned that the dynamic range limitation of the mass spectrometer would exclude the observation of very low-abundance proteins from plasma; in this case, the enrichment step would be mandatory [25]. In addition, the mentioned methods required several steps, such as mixing NPs with proteins and other media, followed by washing out the proteins. In particular, the washing out process could be another factor compromising the experimental accuracy. An incomplete composition of protein corona could be identified due to the incomplete washing-out process, especially for the strongly bound proteins. Furthermore, centrifugation steps are known to perturb the colloidal stability of NPs and, therefore, would induce NP-protein dissociation that would not occur in the biological fluids.

Despite the wide applications of NPs in the biomedical field, the biological interaction of NPs is yet to be fully understood. To make the characterization of the biological interaction of NPs a regular step in the development of NPs it is necessary to develop a convenient analytical tool that provides detailed and reliable information. Herein, surface plasmon resonance imaging (SPRi) is a useful technique in studying the interaction of gold and silver NPs with lipid membranes and proteins. Compared with other available techniques (AFM, mass spectrometry, UV-Vis spectroscopy) that have been used in similar investigations, SPRi can be faster, more accurate and provides a wide range of information in real-time. In fact, SPR has already been explored as a technique to evaluate the adsorption of NPs on proteins biochip or as a complementary technique in the identification of protein corona [21,25,33,34]. However, immobilized NPs were used in these studies, and the proteins were injected onto the modified biochip. Therefore, the colloidal state of the NPs suspension was compromised in comparison with the use of the NPs suspension as it would be injected in vivo.

SPRi is a well-known technique used to monitor the interactions between biomolecules (e.g., proteins, DNA, RNA, peptides) in real-time based on the variation of the refractive index of a prism coated by a thin layer of metal (SPRi biochip) [35,36]. Due to the surface plasmon effect, under the incidence of polarized light at a specific angle on the interface between the biochip surface and a dielectric medium, the energy of the light is almost entirely absorbed by the plasmonic band. Under this condition, an evanescent wave is generated, and the electromagnetic field of a surface plasmon polarization is confined between the boundaries of the metal and the dielectric environment. Its intensity decreases exponentially between these two environments. Small disturbances caused in this field by the adsorption of molecules on the biochip surface are detected as changes in the refractive index.

In the methodology presented herein, lipid membranes and proteins were immobilized on the SPRi biochip surface. The interaction with gold and silver NPs was monitored in real time by injecting these NPs onto the modified biochip surface. Proteins from the blood serum and plasma were chosen since most applications of these NPs predict their intravenous administration. The phospholipid composition of mimetic membranes was varied to evidence the versatility of the technique that allows mimicking the lipid membranes of different cell types, organelles, and cell walls of microorganisms. Therefore, considering that SPRi would allow the evaluation of several different NPs in a fast and standardized way, it could become a convenient step in the design process of NP development for biomedical applications. The combination of the proteins and lipids to be immobilized on the biochip can be chosen according to the target cells and medium, wherein the NPs will be used.

## 2. Results

In this study, the interaction of gold and silver NPs with lipid membranes and proteins from the blood serum and plasma was investigated. For this, two types of biochips were prepared for SPRi assays, as illustrated in Figure 1.

### 2.1. Gold NPs Interaction with Membrane

Three different lipid membranes composed of 1-palmitoyl-2-oleoyl-glycero-3-phosphocholine only (POPC) or combined with cholesterol (POPC/Chol) or dimethyldioctadecylammonium (POPC/DDAB) were constructed on the SPRi biochip. These lipid compositions were chosen aiming to mimic biological membranes and cell membrane domains. POPC is the main component of the cell membrane and bacteria cell wall. Chol is one of the main components of lipid-rafts, which is a type of cell membrane domain that also contains proteoglycans and forms a rigid structure that can flow throughout the cell membrane. Lipid-rafts have been associated with cell uptake mechanisms of NPs. Therefore, the preferential interaction of NPs with this domain may indicate that the NP can internalize into the cells by this domain. The addition of DDAB was a strategy to evaluate the effect of positively charged lipids on the interaction with negatively charged NPs, which could induce cell damage through its positive moieties [37]. Successive injections of NPs at the same concentration were performed on each membrane to evaluate the NPs distribution on the membrane, and the variation of reflectivity was monitored in the function of time (Figure 2). The changes observed after each injection of gold NPs on the three different membranes could be better evaluated by plotting the variation of reflectivity in the function of the number of gold NPs injections (Figure 3). It was observed that the variation of reflectivity increased with the increasing number of NPs injections. For POPC/DDAB membrane, after the fifth injection, the reflectivity slightly changed with additional injections of NPs, indicating the saturation of the membrane. When gold NPs were injected onto the membrane composed of POPC/Chol or POPC, the variation of reflectivity did not reach a plateau, and the reflectivity continued to change with the additional NPs injections, indicating that the NPs continued to interact with the membrane even after the seventh injection (Figure 3).

The different profiles of the interaction of gold NPs could be a result of the different affinity of gold NPs with each type of membrane and because of the different structuring of the membranes. The continuous adsorption of gold NPs on the POPC/Chol or POPC membrane and the low adsorption of gold NPs on the POPC/DDAB membrane are evidenced in Figure 4. The surface concentration density of gold NPs adsorbed on POPC/Chol membrane was (9.0 ± 0.4) mg/mm^2^, whereas the average value on the POPC and POPC/DDAB membranes was (4.0 ± 0.1) and (4.0 ± 0.3) mg/mm^2^, respectively.

The enhanced interaction between gold NPs and the POPC/Chol membrane could be correlated with the results reported previously regarding the cell internalization process of NPs. Cholesterol-rich domains in the cellular membrane, called lipid-rafts, have already been well characterized and mimicked in vitro by assays wherein giant vesicles were used as membrane models [30,31,38,39]. These domains have already been identified as one of the main regions that promote NPs cellular internalization. Preferential adsorption of gold NPs on these domains could lead to the NPs accumulation in a small portion of the membrane, promoting NPs agglomeration and/or aggregation.

In addition to the quantitative analysis of the gold NPs adsorption on lipid membranes of different compositions, it was possible to infer how the adsorption process could occur and if it could be expected the aggregation or agglomeration of NPs on each type of membrane. Figure 5 describes our hypothesis on how the adsorption of gold NPs would proceed on each type of membrane. The adsorption of gold NPs on the membranes containing POPC and POPC/DDAB could be a result of the electrostatic interaction between the negatively charged NPs and positively charged membrane of POPC/DDAB or the positive portion of POPC molecules. However, after the first injections of gold NPs on the membrane, the membrane would be recovered by the NPs, resulting in a negatively charged surface, which hampers the continuous adsorption of gold NPs under the successive injections, and only a low variation of reflectivity could be observed. The adsorption of gold NPs on the POPC/Chol membrane took place on the domains containing cholesterol rather than on the regions of POPC. The preferential and strong adsorption of gold NPs in the cholesterol-rich domains induces aggregation under successive injections of gold NPs on the membrane.

Our hypothesis regarding the adsorption process of gold NPs on the membranes was supported by the differential images generated in the SPRi assay. Figure 6 depicts the images at the end of the SPRi assays after the injection of gold NPs on POPC/DDAB, POPC and POPC/Chol membranes. The clearer areas indicated the regions where the adsorption took place. It was possible to note the homogenous adsorption on the POPC/DDAB and POPC membranes, whereas the image of the POPC/Chol membrane pointed out a different contrast through the surface, which could be indicative of NPs aggregation on the surface.

The different interactions of NPs with membranes of different compositions could lead to different cytotoxic and antimicrobial effects. Several groups have observed different toxicity of NPs in Gram-positive and Gram-negative bacteria, which they attributed to the different lipid composition of the bacteria cell wall [39,40,41,42,43]. Herein, it is shown that by using SPRi, it could be possible to infer the effects of an NP on membranes with varied lipid compositions.

### 2.2. Interaction of NPs with Proteins

In this work, the interaction between gold and silver NPs and several proteins from blood serum was evaluated by SPRi. Even if the composition of the protein corona of these NPs in different mediums has already been reported, there is still a lack of information about the affinity of NPs to different proteins. The application of SPRi in the study of the interaction between NPs and proteins provides information about the extension of the adsorption and also about the affinity between NPs and proteins.

Herein, gold and silver NPs were injected on a biochip functionalized with seven different proteins. The average kinetic curves of variation of the reflectivity in the function of time after injection of gold NPs are presented in Figure 7 for each kind of protein. As can be seen, the injection of gold NPs on proteins biochip resulted in different values of variation of reflectivity, depending on the identity of the protein. This result indicated that the extension of adsorption of gold NPs changed according to each blood protein immobilized on the biochip. Therefore, a different affinity of gold NPs to each protein would be expected.

The same assay was repeated with silver NPs. The analysis of the kinetic curves allowed us to infer the affinity of each protein with gold and silver NPs. The proteins with high affinity should bind to NPs more strongly than the proteins with low affinity. Therefore, after the injection of NPs (step I; Figure 7), a variation of reflectivity was observed when they bound to the proteins on the biochip surface (association phase shown in step II; Figure 7). Following this, the reflectivity decreased due to the washing out process by the running buffer (dissociation phase shown in step III; Figure 7). Finally, a plateau was observed that indicated the quantity of the bound NPs to the protein on the biochip.

Figure 8 shows the average values and standard deviation of the variation of reflectivity after the injection of gold (Figure 8a) and silver NPs (Figure 8b) on each kind of protein. These values were calculated by subtracting the initial value of reflectivity of the baseline from the value of reflectivity after the dissociation phase. The evaluation of the variation of reflectivity provides insight into the extension of the adsorption of NPs on each type of protein. The higher the adsorption, the higher the affinity between the protein and NPs.

It was possible to observe that among the tested blood serum proteins, fibrinogen, ApoA and IgG were the proteins that interacted more with gold and silver NPs. In the assays performed with gold NPs, the average value of the variation of reflectivity was (10 ± 1)%; (9.2 ± 0.5)%; (7.3 ± 0.5)% and (5.6 ± 0.8)% on the spots of fibrinogen, IgGR, ApoA and IgGH respectively, whereas on the spots of HSA and BSA, the average values were six times lower (1.6 ± 0.3)%. The difference in the variation of reflectivity between the spots of IgGR and IgGH pointed out the higher affinity of gold NPs to the animal protein compared with the human protein.

In the assays performed with silver NPs, the highest average value of the variation of reflectivity was observed on IgGR spots (2.1 ± 0.2)%, followed by ApoA (1.7 ± 0.1)% and fibrinogen (1.1 ± 0.1)%. Different from what was observed with the gold NPs, the interaction of silver NPs did not show a high variation between the tested proteins. The interaction of silver NPs with the other proteins resulted in values of variation of reflectivity only slightly lower in the 0.7–0.9% range.

In addition to the information regarding the composition of the NPs protein corona and the comparative evaluation of proteins adsorption on the NPs surface, the SPRi technique can also be applied to quantitatively evaluate the affinity between them. The kinetic curves obtained in the assays with gold and silver NPs were used to calculate the dissociation constant (k_off_) after the interaction of NPs with proteins (see Appendix A for a detailed description of the calculus). The values are depicted in Table 1.

The values of k_off_ reflected the same preferential interaction indicated by the values of the variation of reflectivity. Since the k_off_ refers to dissociation, the lower its value, the stronger the affinity between the NPs and proteins. Therefore, the lowest values of k_off_ amongst the interaction of gold NPs were observed with fibrinogen and IgGR (k_off_ = 1.1 × 10^−3^∙s^−1^). Notably, this value is at the same order of magnitude as the strong interaction between transferrin receptor:T7 peptide (k_off_ = 6.8 × 10^−4^∙s^−1^) [44] and flavoenzyme:NADP+ cofactor (k_off_ = 2.0 × 10^−2^∙s^−1^) [45]. Compared with the values found for silver NPs, it was possible to observe that despite the lower adsorption of these NPs on the BSA and HSA, the k_off_ was very similar among all the tested proteins, indicating the similar affinity between these proteins and silver NPs. In this case, the composition of the protein corona of silver NPs would be expected to depend mainly on the concentration of the proteins in the medium, which could change with time or with the conditions of the medium. In contrast, the composition of the protein corona of gold NPs would be determined by the concentration and the affinity between gold NPs and each protein.

In contrast with the numerous works that reported on the composition of protein corona formed on NPs, works reporting the quantitative data of affinity between proteins and NPs are much less abundant. In the work of Gorshkov et al. [25]., the binding affinity of different proteins and silver NPs (citrate capped NPs of 60 nm) was inferred from the data obtained at different temperatures. In this case, the proteins that remained bound to the NP even after the temperature rise were considered strongly bound. Differently from what we observed here, their studies pointed out that protein binding on silver NPs was selective and that the protein corona formed in the blood would contain only a very small subset of the entire plasma proteome. Their work included about 300 proteins, and the persistent protein corona did not include serum albumin, apolipoprotein and immunoglobulins. Notably, the silver NPs investigated here were smaller (30 nm), and the assays were performed only at 37 °C. Lai et al. [46] also reported the low content of albumin in the protein corona of 20 nm silver and gold NPs. The authors observed the preferential binding of fibrinogen on gold NPs proteins and the low abundance of albumin and globulins on both gold and silver NPs protein corona. In the work of Patra et al. [47], the composition of protein corona and the affinity between plasma protein and gold NPs were evaluated by using multiplexed surface plasmon resonance. The authors found similar values of k_off_ in the interaction with IgGR, fibrinogen, ApoA and IgGH (Table 1). Still, the value of k_off_ found here in the interaction with has was higher (5.50 × 10^−3^∙s^−1^) than the value reported by them (1.8 × 10^−3^∙s^−1^) [47]. This difference may arise from the fact that in the work of Patra et al., the gold NPs were immobilized on the biochip surface, whereas in our assays, the gold NPs were in suspension. Once their immobilization hampered the movement of NPs, the dissociation between them and the protein was slower, and therefore, the k_off_ was lower.

The application of SPRi in the study of proteins association to and dissociation from NPs has also been explored before with non-metallic NPs [33,34]. Inversely to what was done herein, the NPs were immobilized on the surface of the biochip, and the proteins from plasma were injected onto the biochip surface. Cedervall et al. used SPR only as a complementary technique to characterize the protein corona formed on 70 nm and 200 nm copolymeric NPs [33]. Although they have quantified exchange rates and binding affinities of proteins to NPs, their approach did not lay only on the SPR data but also on the data obtained by isothermal titration calorimetry, SDS-PAGE and liquid chromatography. From the SPR assay, they concluded that when the whole plasma was injected onto the NPs, more than one protein was associated with NPs, and to calculate dissociation constants, the SPR data were adjusted by considering two simultaneous association-dissociation events. The calculated dissociation rates revealed a faster dissociation of proteins fibrinogen and HSA from hydrophobic NPs than from hydrophilic NPs. Similar to what was observed here and in the aforementioned works, the affinity of proteins to NPs depends on the NPs surface and the identity of the protein.

#### Interaction between Gold NPs and Proteins Studied by Localized Surface Plasmon Resonance

Gold and silver NPs can also be used as a sensitive tool for bio-sensing based on localized surface plasmon resonance (LSPR). Both local refractive index changes, for example, induced by the coating of chemical/biochemical molecules and self-assembly of the nanoparticles due to aggregation, can lead to some spectral shifts of the LSPR extinction peak. In this study, we also utilized these phenomena to develop a very simple and straightforward approach for the evaluation of the interaction between NPs and proteins in solution.

Gold NPs have a characteristic absorbance band in the region of 500 nm, whereas silver NPs have a plasmon band around 400 nm. The values of λmax vary according to the size of the NPs. For example, the 20 nm gold NPs used in this work presented λmax at 525 nm and the silver NPs at 431 nm. When the optical properties of these NPs change due to aggregation or the coating by chemical/biochemical molecules, the plasmon band becomes wider and shifts to higher wavelengths. Herein, the interaction of gold and silver NPs with serum blood proteins was also monitored by UV-Vis spectroscopy in order to compare with the SPRi results.

The spectra were obtained immediately after mixing the proteins with the NPs and 4 h after incubation. The spectra of gold NPs are depicted in Figure 9. Similar behavior was observed with the silver NPs. It was possible to note there was a red shift of the spectrum after the interaction between NPs and proteins. For easier comparison purposes, the values of λmax and absorbance of each spectrum are depicted in Table 2. Variations of absorbance and λmax shifting were evaluated by comparing the spectra of gold and silver NPs mixed with proteins with the spectra of NPs stock suspension and of the NPs in HEPES buffer solution. Once no changes had been observed in the spectrum of NPs suspended in HEPES solution in comparison with the spectra of NPs stock suspension, it was possible to conclude that the changes observed in the spectra of NPs mixed with the proteins were a consequence of the interaction between them.

Immediately after mixing NPs and proteins, only slight shifts of plasmon band were observed. The most prominent red shift was observed in the spectra of gold NPs in the presence of fibrinogen, indicating that the gold NPs interacted more with fibrinogen than with the other proteins. After 4 h of incubation, the spectra of gold NPs incubated with fibrinogen, ApoA and IgGR, became wider, and a significant shift of the plasmon band to higher wavelengths at the region of 570 nm was observed. Nevertheless, no difference was observed in the spectra of gold NPs incubated with transferrin, BSA, HSA and IgGH. These results suggest that subtle changes on the NPs surface, such as due to low interaction of proteins, could not be detected by UV-Vis spectroscopy. In this case, SPRi was advantageous since the interaction between the gold NPs and all the tested proteins was detected and monitored in real time.

The results found with silver NPs were also in agreement with the results from the SPRi assay. The most prominent shifts towards higher wavelengths were observed with the fibrinogen and ApoA proteins. In addition, as also pointed out by the SPRi results, the silver NPs showed a similar interaction with all the studied proteins since the λmax shifting was not very different among the absorption spectra.

The difference between the results observed immediately after mixing gold and silver NPs with proteins and after 4 h of incubation suggested that the interaction with the proteins increased with time. The decrease in absorbance was probable a result of NPs aggregation since the proteins with higher affinity with gold NPs could coat all the NPs surfaces, decreasing the surface charge. Although a similar trend was observed in the UV-Vis and SPRi results regarding which proteins would interact more with gold and silver NPs, the UV-Vis results were less informative and less sensitive regarding the kinetics and affinity differences.

## 3. Discussion

The biological interaction of NPs has been the subject of extensive research in the last few years. In particular, the interaction of NPs with lipid membranes and proteins has attracted more attention due to their impact on the circulation time, targeting activity, cell internalization and therapeutic efficiency of NPs. Several theoretical and experimental approaches have been developed to fully characterize the interaction of NPs with lipid membranes and proteins regarding their kinetic, mechanism and composition [18,21,48,49]. Herein, we report on the use of SPRi assays aiming at a fast and accurate characterization of NPs protein corona and interaction with lipid membranes.

Although the pathways for NP’s cytotoxicity are diverse and dependent upon the nature of the NP and the biochemical environment, numerous studies have provided evidence that the direct contact between NP and cell membranes leads to cell inactivation or damage and may be a primary mechanism for cytotoxicity [29,30,50,51,52]. Furthermore, the interaction between NPs and lipid membranes can also suggest a potential antimicrobial activity of the NPs. In fact, the effect of gold NPs on the outer membrane of the bacteria cell wall has evidenced their potential application as antimicrobial agents [43,51,53,54,55]. Our findings have shown that it is possible to easily and quickly evaluate the interaction between gold NPs with membranes of different compositions using SPRi. The observed low affinity of gold NPs with POPC membranes was in line with the observed affinity of negatively charged NPs with POPC membranes or membranes with a similar lipid composition. The crucial role of the electrostatic interaction between NPs and membranes has been evidenced by several researchers [12,56,57,58]. Hong et al. [59] pointed out that the adsorption of NPs to the membrane was a process driven by electrostatic interaction, which induced an osmotic imbalance in the membrane. In the work of Katz et al., the authors could prove that an amphiphilic monolayer created on the NPs surface inhibited the interaction of gold NPs with the plasma cell membrane, and no evidence of membrane poration or cell death was observed [60]. On the other hand, by using gold NPs with different charges and surface densities, Lin et al. [30] could observe that the NPs spontaneously adsorbed on the surface of the membrane or penetrated the bilayer. Although it is expected that the surface of the POPC/DDAB membrane has a slightly positive charge, the DDAB molecules would be homogeneously spread throughout the membrane, and it was not enough to increase the adsorption of gold NPs on the membrane in comparison with the membrane containing only POPC. Therefore, the NPs were homogeneously distributed on the membrane surface and showed saturation after successive injections.

The continuous adsorption of gold NPs on the POPC/chol membrane could be explained by the preferential interaction of these NPs with cholesterol-rich domains of membranes. This interaction could be promoted by the instability of the cholesterol−cholesterol interaction in the POPC membrane. As reported by Dai et al. [61], cholesterol clusters can pass through a conformational change to avoid water penetration between cholesterol headgroups. Then, the presence of gold NPs would promote the preferential interaction between cholesterol monomers than cholesterol-cholesterol interaction in the cluster. Under their strong interaction, subsequential gold NPs injected onto the membrane would promote NPs agglomeration/aggregation with the gold NPs previously adsorbed. The agglomeration/aggregation of NPs will result in bigger structures, and as already evidenced before, changes in size will result in different interactions with cell membranes [60,62,63,64]. The effect of NPs size on the interaction with membrane has also been pointed out to change the antimicrobial activity of NPs. In the work of Hayden et al. (2012), hydrophobic cationic 2 nm gold NPs were able to lyse the *B. subtilis* cell wall, whereas 6 nm NPs did not cause any toxic effect. The authors associated this result with the different clustering of 2 nm and 6 nm NPs on the bacteria’s outer membrane. The 2 nm NPs formed big aggregates after adsorption on the cell wall, whereas the 6 nm NPs were homogeneously distributed on the cell wall and formed small aggregates [42].

Once most of the NPs under development has been focused on biomedical application, the protein corona formed on different types of NPs has been mainly investigated in the blood serum and plasma. The results showed that the protein corona composition could be influenced by the core, surface charge and surface chemistry of the NPs [17,27,33,46]. It could also be changed by the conditions of the biological medium, such as concentration of proteins, pH and temperature [6,25,27]. The strategy addressed by most of the researchers has been based on the indirect measurement and other procedures that included the centrifugation of NPs after their incubation with the biological medium and quantitative analysis of the proteins in the supernatant. However, centrifugation is known to induce NPs aggregation and perturb the association of biomolecules adsorbed on the NPs surface. Therefore, it would compromise the accuracy of these assays.

Herein, the interaction of silver and gold NPs with proteins was evaluated by SPRi. The identification of ApoA, fibrinogen and immunoglobulins as the proteins that gold and silver NPs would have a higher affinity for can be considered as an indication that these proteins would be the major components of protein-corona formed on the NPs surface when these NPs would be in blood serum (and/or) plasma. This observation is in line with several works reported before [6,46,47,65]. Chan et al. [6] have reported on the composition of the protein corona formed on gold NPs. The authors investigated the effect of the NPs size and surface chemistry on the protein corona composition. They also showed how the composition changed with time and with media composition and cell phenotype. The composition of protein corona of 15 nm citrate-stabilized gold NPs was characterized by liquid chromatography-tandem mass spectrometry (LC-MS/MS) after incubation with a culture medium conditioned by A459 cells which contained 581 proteins. The group of proteins that composed the majority of the protein corona changed with time. Although they have not identified fibrinogen, IGR or ApoA, they reported the presence of APO E, fibronectin, Beta-2 glycoprotein and thrombospondin-1 as the main components of the protein corona formed right after the incubation. In the work of Dobrovolskaia et al. [65], the protein corona formed on 30 nm gold NPs when these NPs were in contact with human serum was characterized by 2D page and mass spectrometry. They have found that the main components of protein corona were fibrinogen, albumin and ApoA. In the work of Gorshkov et al. [25], ApoA also appeared amongst the most abundant proteins in protein corona formed on 60 nm citrate-stabilized silver NPs. Interestingly, ApoA was also one of the main proteins of corona formed on metal oxide and liposomes NPs. In the protein corona formed on 50 nm silica NPs, besides ApoA, the authors also identified the presence of other apolipoproteins, such as ApoB and ApoE [22]. Similarly, Yang et al. [66] indicated APO as one of the main components of protein corona formed on liposomes of different ratios of negatively charged and zwitterionic lipids. ApoC was the most abundant protein in the protein corona of liposomes composed of negatively charged lipids. Gorshkov et al. [25] reported on the selective interaction of silver NPs with plasma proteins resulting in a protein corona that included only a very small subset of the entire plasma proteome. Nevertheless, they adverted that due to the dynamic range limitation of the mass spectrometer, the observation of very low-abundance proteins from plasma samples could be compromised.

In comparison with the techniques already used to identify the NPs protein corona, mainly mass spectrometry, liquid chromatography and SDS-PAGE, the present work highlights the application of SPRi as a convenient and systematic technique to investigate NP–protein interaction. Most important, the informative aspect of SPRi configures its main advantage over the other available techniques. Through SPRi, it was possible not only to identify which proteins would be the most abundant in the protein corona formed on each type of NPs but also to perform a comparative investigation of the interaction of NP with several types of proteins at the same time. SPRi was notoriously shown to be a straightforward method that allows the calculation of dissociation constant of NPs-protein interaction with high accuracy, which most researchers pointed out as the main difficulty faced by the methods they used. Nevertheless, one of the drawbacks of SPRi is the limitation regarding NPs size under investigation since the sensitivity decreases with the distance from the biochip surface due to the evanescent plasmon field, and for this reason, the sensitivity to particles or aggregates larger than 300 nm would be compromised. Even so, SPRi has been very useful in evaluating the interaction of NPs < 100 nm. In our previous work, the SPRi was successfully applied in the investigation of the interaction of 30 nm NPs with membranes. In this case, SPRi allowed us to investigate how lipid coating changed the interaction of NPs with lipid membranes of different surface charges [63]. Similarly, some other groups applied SPR in order to evaluate the association of proteins with NPs [33,34,47]. Cedervall et al. [33] investigated the adsorption of fibrinogen and HSA to 70 nm NPs composed of copolymers with different hydrophobicity [33]. Herein, SPRi was successfully applied in the study of the interaction of 30 nm silver and 20 nm gold NPs with proteins and membranes.

Since protein corona changes NPs colloidal stability and interaction with cells, the prediction of its composition would be a crucial step in the NPs design and development process. The influence of NPs surface charge and chemistry on the formation of protein corona brought to light how NPs coating could be used to control protein corona composition. The results obtained herein showed that SPRi would be a simple method to evaluate the performance of different NPs coatings as modulators of NPs interaction with proteins and lipid membranes. Innovative coatings could be developed in order to obtain a fine tune of the protein corona composition and cell targeting. Yang et al. [66] showed that protein corona composition could be modulated by tuning the surface charge of liposomes. Liposomes composed of higher ratios of negatively charged lipids to zwitterionic lipids recruited higher concentration of apolipoproteins (APOC, APOA-1), whereas liposomes composed mainly of 1,2-dioleoyl-sn-glycero-3-phosphocholine (DOPC) showed protein corona with a higher content of immunoglobulins such as IGKC and IGMH. Following what was done in our previous work [63], the lipid coating of NPs with the different phospholipids could also be a strategy to modulate the composition of the protein corona. Another candidate for a stable NPs coating could be composed of proteins with high affinity to NPs surfaces. In this case, the SPRi assays would be helpful in the identification of proteins that would strongly bind to the NPs surface, hampering the exchange process once they are found in a biological medium. Moreover, to control the protein composition of the protein corona, Mosquera et al. [27] presented an ingenious approach to induce reversible disruption of the protein corona by using external stimuli based on the host-guest interactions. Their results reinforced the changes in the protein corona composition with the conditions of the external medium.

In this context, SPRi would also be a convenient technique to investigate how NP-protein interaction would be affected by NPs coating and environmental conditions in function of time. It is important to remember that the environmental conditions would change through the NPs pathway in the body, as they would pass from the blood to several tissues. In addition, biological environments have a dynamic composition that responds to metabolism and diseases. Moreover, one should also consider that the contrary could be true, which means the presence of NPs and the way they interact with cells, especially with the cell membrane, could alter the cell metabolism and, therefore, change the proteins and metabolites released into the extracellular environment. Considering the complexity of NPs interaction with protein and membranes and their effects on NPs behavior in vivo, the development and design process of the NPs should count on the SPRi as a straightforward and systematic method to provide accurate information regarding their biological interaction. SPRi assays could also be combined with other techniques, enriching the information regarding the biological behavior of NPs. For example, biomolecular interactions can be further investigated by bioluminescence resonance energy transfer (BRET) [67], fluorescence cross-correlation spectroscopy [68] and by atomic force microscopy to quantitatively assess the biomolecular dynamics and molecular recognition imaging (MRI) [69].

In order to summarize the idea, Figure 10 depicts a flowchart of how SPRi assays could be integrated into the development process of NPs for biomedical applications. Normally, aiming for certain applications, the NPs are designed to present suitable size, morphology, surface charge and colloidal stability. Following the synthesis, these NPs are already applied in vitro, which consists of time-consuming and expensive assays (Figure 10; step I). Notably, once the in vitro results reveal the unsatisfactory performance of NPs, the process must return to synthesis. However, by using SPRi (Figure 10; step II), in vitro results could already be predictable by evaluating the biointeractions of NPs as soon as NPs synthesis is finished.

## 4. Materials and Methods

### 4.1. SPRi Assay with Membranes

Biomimetic membranes were prepared on SPRi biochip according to the method previously described [63]. The following sections briefly describe the method with some modifications. 

#### 4.1.1. Preparation of the Biochip

Before the assay, the SPRi prism (SPRi-Biochips™; Horiba) was cleaned with sulfuric acid: hydrogen peroxide (4:1) solution. After washing with water and drying with N_2_, the SPRi prism was immersed in a 10 mM 1-dodenacothiol (Sigma-Aldrich) ethanolic solution for 12 h. Finally, the functionalized SPRi prism was thoroughly washed with ethanol and dried with N_2_.

#### 4.1.2. Vesicles Preparation

The vesicles were prepared by drying a solution of lipids in chloroform under a flow of N_2_, followed by solubilization of the lipidic film with warm (36 °C) PBS buffer. All the lipids were acquired from Avanti Polar Lipids. The total lipid concentration was 6.6 mM. In the POPC: Cholesterol, the molar ratio was 1:1, and the POPC/DDAB vesicles were prepared with 98% of POPC and 2% of DDAB. Then the vesicle preparation was completed by a freeze-tooling process. After about 6 repetitions of the freeze-heating process, the suspension of vesicles was extruded about 11 times in an Avanti polar Lipids extruder with a Millipore membrane (0.1 µm). The vesicles were characterized regarding size and Zeta potential by dynamic light scattering (Delsa Nano C, Beckman Coulter, Brea, CA, USA).

#### 4.1.3. SPRi Assay

In the SPRi assay, several spots were chosen to analyze the whole surface of the biochip. At the beginning of the experiment, a solution of SDS 1% was injected to clean the surface of the biochip and to wash off the excess dodecanethiol. After stabilizing the kinetic curves, the vesicle suspension was injected to form the membrane on the biochip surface. After the stabilization, successive injections of gold NPs were performed to evaluate the adsorption of the NP and saturation of the membrane.

### 4.2. SPRi Assay with Proteins

#### 4.2.1. Preparation of the Protein and Prism

All the SPRi assays presented in this work were performed using SPRi Lab+ equipment (Horiba, France). At first, the SPRi prism (SPRi-Biochips™; Horiba, France) was cleaned by a plasma cleaning in a Femto plasma cleaner (Diener Electronic, Germany) (0.6 mbar, 75% Oxygen, 25% Argon, power 40 W, 3 min) to ensure that the surface was completely free of any contaminants. Following that, the SPRi prism was immersed in a solution of N-hydroxysuccinimidyl 11-mercaptoundecanoate (Thiol-NHS; Sigma-Aldrich, France) (1 mM in ethanol) overnight. In such a way, the gold surface on the prism was covered by the self-assembled monolayer of thiols with active terminal groups for binding the proteins on the chip surface. Then, the protein solutions were spotted on the SPRi prism using a non-contact spotter (sciFLEXARRAYER S3, SCIENION, Dortmund, Germany). This system enables many spots to be made on the same surface with a low volume of solution (about 100 nL for drop). The following proteins were used: IgG from human serum, human serum albumin (HSA), apolipoprotein-A (Apo-A), fibrinogen from human plasma, and transferrin. Two animal proteins were used for comparative purposes: bovine serum albumin (BSA) and IgG from a rabbit. All of them were purchased from Sigma-Aldrich (France). The solutions of these proteins were all prepared at the concentration of 0.01 mg/mL in phosphate buffer solution (PBS) with pH 8.0. 2 h after spotting, the SPRi biochip was washed carefully with PBS. Finally, the NHS-containing active surface between protein spots on the SPRi biochip was blocked with ethanolamine (Sigma-Aldrich, France) at 1 M in PBS pH 8.0 for 1 h. After washing, the SPRi biochip was stored in HEPES buffer solution at 4 °C.

#### 4.2.2. SPRi Experiment Protocol

The SPRi biochip containing the proteins array was set up in Horiba SPRi (Lab^+^) equipment at 37 °C. All the spots were labeled according to the respective protein. After the stabilization of the kinetic curves by using HEPES (pH = 7.4) as a running solution, the NPs were then injected. The gold NPs of 20 nm at the concentration of 7.00 × 10^10^ NPs/mL were purchased from BBI Solutions, and silver NPs were previously synthesized and characterized by transmission electron microscopy as having 30 ± 8 nm of diameter and at the concentration of 1.46 × 10^11^ NPs/mL [16].

### 4.3. Data Analysis

Data obtained from SPR measurements were analyzed using the Origin 2018 and Matlab R2020a software. Each injection results in an interaction between the sample and the various points previously chosen on the sensor surface. The reflectivity variation values were obtained considering the interaction of at least 10 points on the sensor surface for each type of immobilized molecule. Therefore, it was possible to represent the variation values by the mean and the standard deviation. Finally, the average values of reflectivity variation were used to obtain the surface concentration density. The dissociation step represented in the kinetic curve was considered to obtain the k_off_ constant. All calculations are described in the Appendix A.

## 5. Conclusions

The valuable application of SPRi in the characterization of two of the most important interaction of NPs (membranes and proteins) has been shown. It is expected that this work would motivate the application of SPRi in the most diverse investigation and design process of new NPs for biomedical applications. The authors highlight that there is still a lot to be explored in this subject. For example, the SPRi technique also provides a simple way to investigate the interaction at different conditions such as different protein concentrations, medium composition (e.g., presence of ions) and temperature without requiring additional experimental steps. In this case, it would only require setting up the conditions of the running buffer or in the biochip functionalization. Therefore, SPRi will provide much more information in the field of nanotoxicology and nanomedicine.

## Figures and Tables

**Figure 1 ijms-24-00591-f001:**
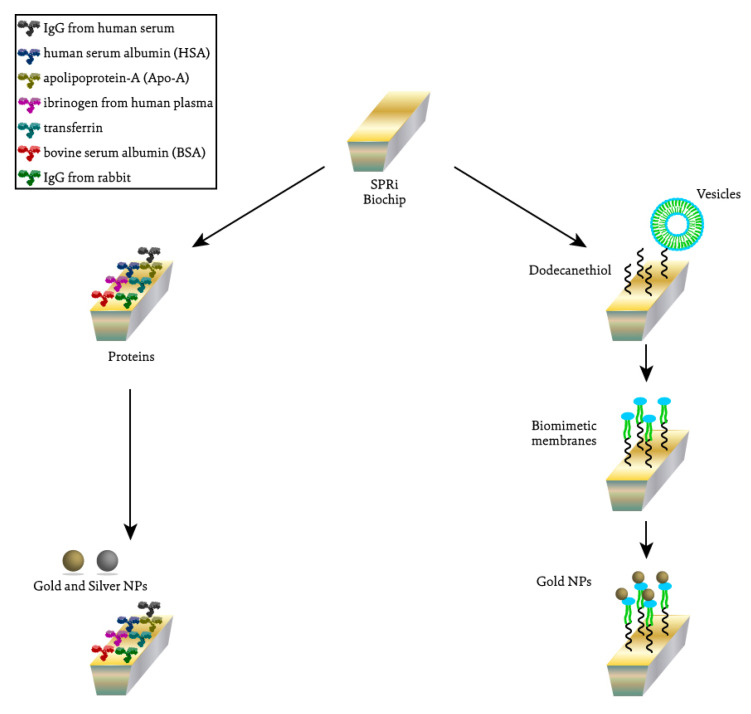
Schematic representation of the different SPRi assays performed to evaluate the interaction between silver and gold NPs and proteins (**on the left**) and between gold NPs and lipidic membranes (**on the right**).

**Figure 2 ijms-24-00591-f002:**
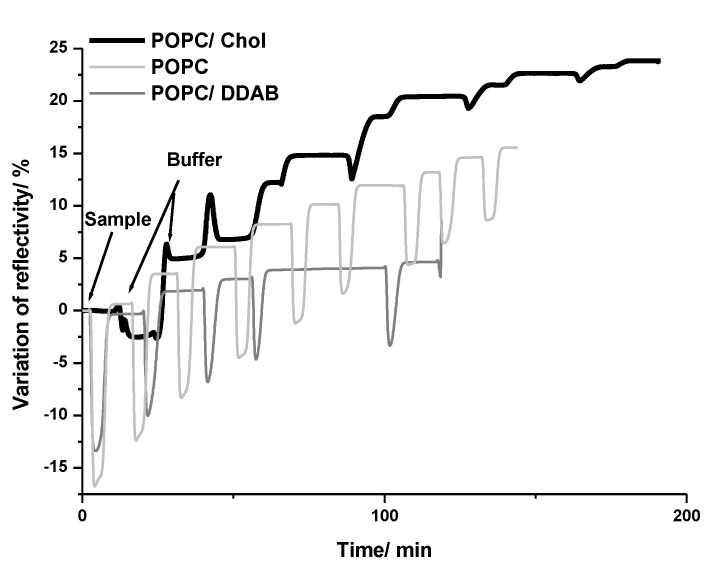
Average kinetic curves variation of reflectivity on the biochip after injections of gold NPs on POPC, POPC/DDAB and POPC/Chol membranes.

**Figure 3 ijms-24-00591-f003:**
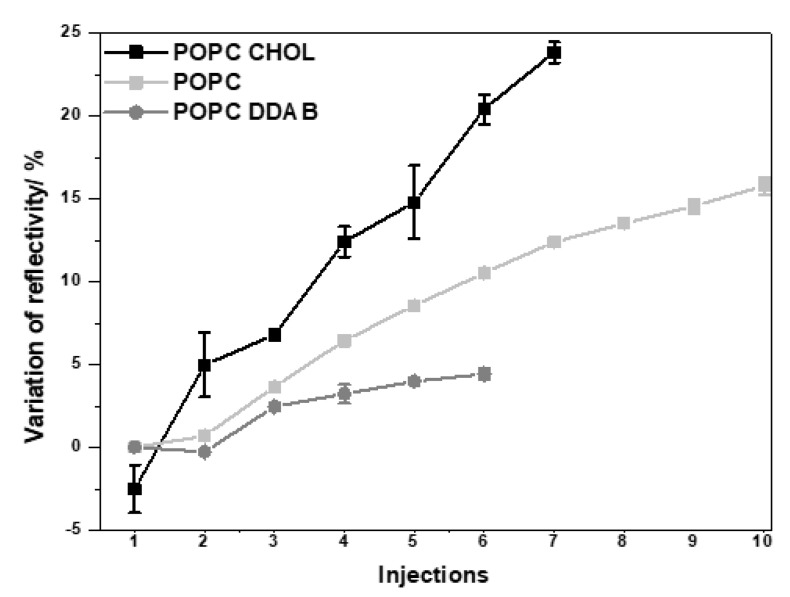
Variation of reflectivity in the function of the number of injections of gold NPs on POPC, POPC/DAB and PC/Chol membranes.

**Figure 4 ijms-24-00591-f004:**
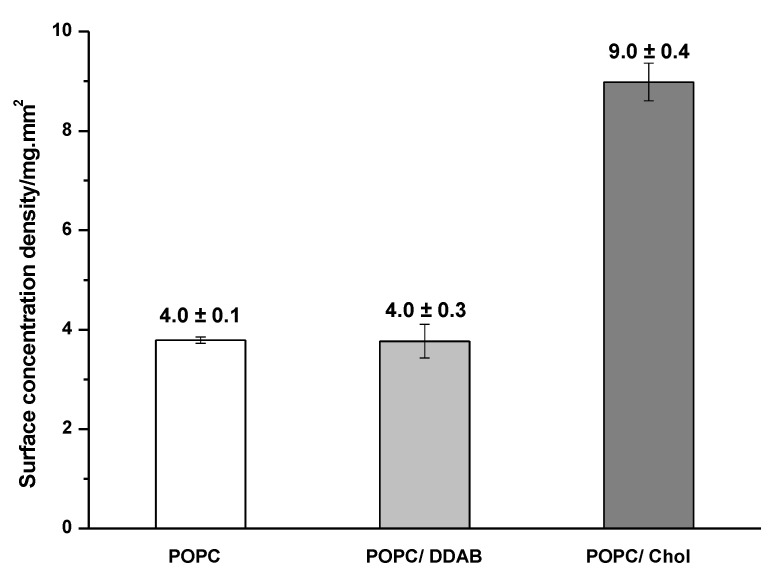
Surface concentration density of gold NPs adsorbed on the membrane of each composition after the SPRi assay was completed.

**Figure 5 ijms-24-00591-f005:**
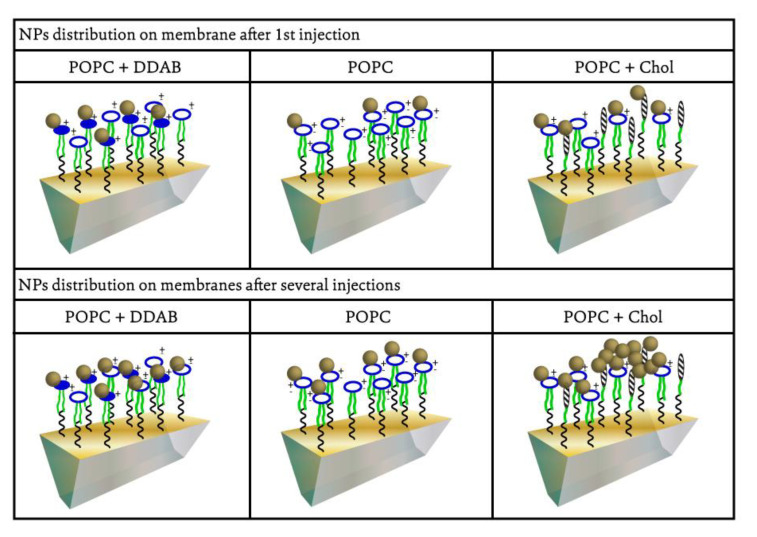
Schematic representation of the gold NPs adsorption on lipid membranes of different compositions under successive injections on SPRi biochip. Gold NPs (golden spheres) are homogeneously adsorbed on POPOC (white/blue dots) and POPC/DDAB (blue dots) membranes. The preferential binding to the region with cholesterol (hatched dots) promote gold NPs aggregation after several injections on the biochip.

**Figure 6 ijms-24-00591-f006:**
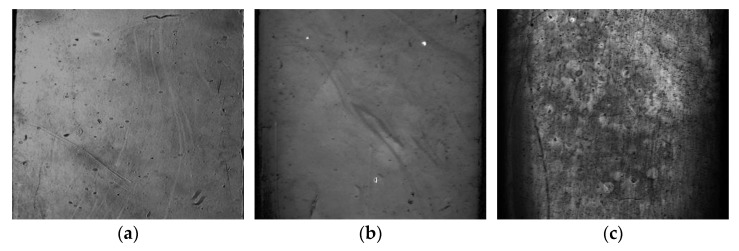
Differential images of the biochip surface after SPRi assays. (**a**) Gold NPs injected onto the POPC/DDAB membrane. (**b**) Gold NPs injected onto the POPC membrane. (**c**) Gold NPs injected onto the POPC/Chol membrane.

**Figure 7 ijms-24-00591-f007:**
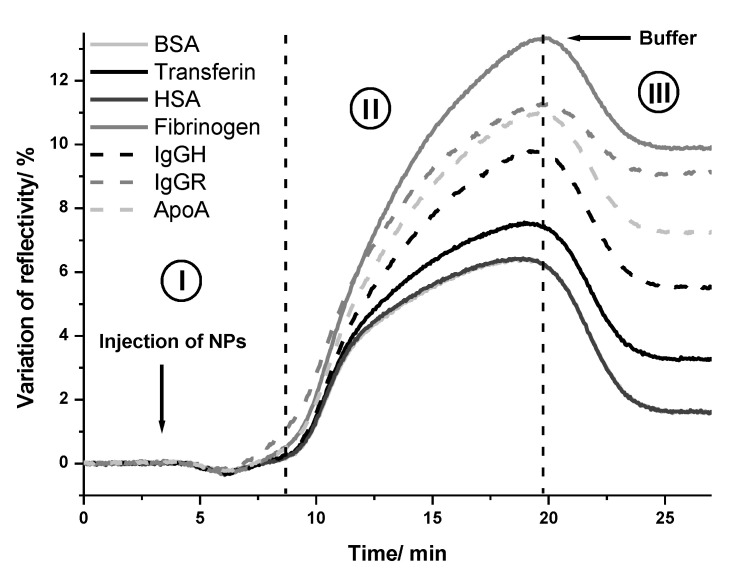
Kinetic curves of the interaction between gold NPs and proteins. Average of the variation of reflectivity on the protein spots under gold NPs injection in the SPRi.

**Figure 8 ijms-24-00591-f008:**
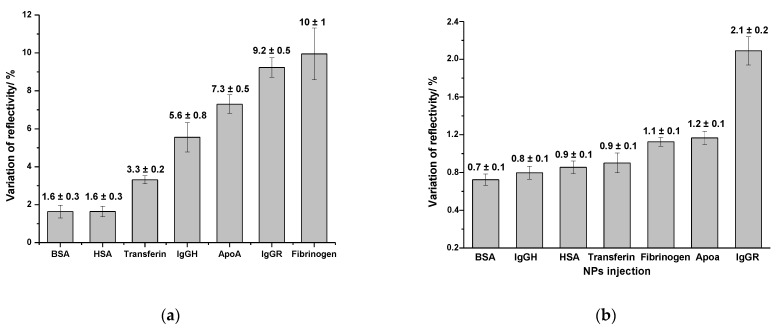
Average and standard deviation of the variation of reflectivity observed in the SPRi curves after the injection of gold NPs (**a**) and silver NPs (**b**) on the protein-functionalized biochip surface.

**Figure 9 ijms-24-00591-f009:**
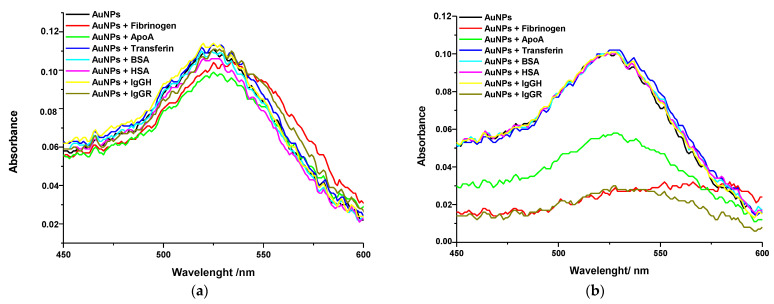
UV-Vis spectra of gold NPs in the presence of proteins (**a**) immediately after mixing and (**b**) after 4 h of incubation.

**Figure 10 ijms-24-00591-f010:**
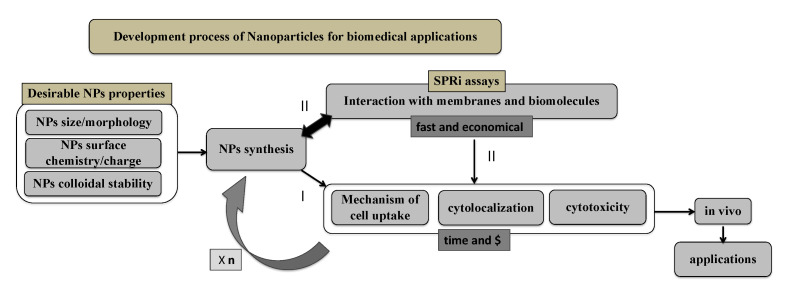
Flowchart of an example of how the SPRi assays could be incorporated in the development process of NPs for biomedical applications providing fast and less expensive assays to predict in vitro results. As soon as NPs synthesis is finished, their biological interactions could be evaluated by SPRi (step II) instead of following it directly to in vitro assays (step I) which are time-consuming and expensive. SPRi assays are faster and of low cost. During the NPs development process, the synthesis has to be adjusted “*n*” times after the evaluation of in vitro results. SPRi assays provide a much more practical technique to adjust the synthesis and biological behavior of NPs.

**Table 1 ijms-24-00591-t001:** Values of k_off_ calculated by using the SPRi kinetics data of the interaction between AuNPs, AgNPs and proteins, and the values of K_off_ reported in the indicated references. The color is used to differentiate the animal proteins (green) from human proteins (blue).

Protein	k_off_ (AuNPs)/s^−1^	k_off_ (AgNPs)/s^−1^	k_off_ (AuNPs) [25]
IgGR	(1.10 ± 0.02) × 10^−3^	(1.50 ± 0.01) × 10^−3^	2.3 × 10^−3^∙s^−1^
Fibrinogen	(1.40 ± 0.02) × 10^−3^	(1.60 ± 0.01) × 10^−3^	2.0 × 10^−3^∙s^−1^
ApoA	(1.90 ± 0.02) ×10^−3^	(1.40 ± 0.01) × 10^−3^	1.6 × 10^−3^∙s^−1^
IgGH	(2.60 ± 0.04) × 10^−3^	(1.40 ± 0.01) × 10^−3^	2.3 × 10^−3^∙s^−1^
Transferrin	(3.60 ± 0.05) × 10^−3^ s^−1^	(1.50 ± 0.01) × 10^−3^	
HSA	(5.50 ± 0.01) × 10^−3^ s^−1^	(1.60 ± 0.01) × 10^−3^	1.8 × 10^−3^
BSA	(5.50 ± 0.01) × 10^−3^ s^−1^	(1.30 ± 0.01) × 10^−3^	

**Table 2 ijms-24-00591-t002:** Values and variations of λmax and absorbance of gold and silver NPs immediately after mixing and after 4 h of incubation with proteins.

Sample	Immediately		After 4 h		Variation	
	λmax (nm)	Absorbance	λmax (nm)	Absorbance	Δλmax (nm)	ΔAbsorbance
			Gold NPs			
Gold NPs	525	0.12	528	0.12	3	−0.01
NPs + HEPES	525	0.11	528	0.10	3	−0.01
NPs + Fibr	534	0.10	584	0.04	50	−0.07
NPs + ApoA	525	0.10	529	0.06	4	−0.04
NPs + Transf	529	0.11	525	0.10	−4	−0.01
NPs + BSA	519	0.11	526	0.10	7	−0.01
NPs + IgGH	520	0.11	524	0.10	4	−0.01
NPs + IgGR	528	0.11	529	0.09	1	−0.02
			Silver NPs			
Silver NPs	431	0.09	428	0.11	−3	0.01
NPs + HEPES	412	0.14	430	0.10	18	−0.03
NPs + Fibr	415	0.14	435	0.05	20	−0.09
NPs + ApoA	389	0.15	410	0.12	21	−0.02
NPs + Transf	408	0.17	415	0.14	7	−0.03
NPs + BSA	411	0.15	430	0.09	19	−0.06
NPs + IgGH	408	0.16	428	0.07	20	−0.09
NPs + IgGR	408	0.14	429	0.11	21	−0.04

## Data Availability

Raw data were generated at the ICT-UNIFESP and UMC. Derived data supporting the findings of this study are available from the corresponding authors D.B.T. and D.C.A., upon request.

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
