# Peer review of "Interaction between Nanoparticles, Membranes and Proteins: A Surface Plasmon Resonance Study"

_ijms, 2022, doi:10.3390/ijms24010591_

Round 1
Reviewer 1 Report
The manuscript entitled "Interaction between Nanoparticles, Membranes and Proteins A Surface Plasmon Resonance Study" publishes the results of a study of the interaction of gold and silver nanoparticles with proteins and lipid membranes, performed mainly using the SPRi method. The article is written competently, the narrative is consistent, the conclusions are clear. It is proposed to accept the article after minor changes.
1) Lines 39-41. A cumbersome written sentence, it is desirable to reformulate it, to facilitate perception.
2) Lines 48-57. In this paragraph, we can also mention that research has been conducted on the creation of composite materials from nanoparticles and organic molecules, for example, complexes of magnetic nanoparticles and DNA. https://doi.org/10.3390/polym14020344
3) Figure 1 should preferably be presented in good quality.
4) It is advisable to briefly describe the difference between the Figure 2 and Figure 3 construction method. These graphs are similar in observed effects and have the same axis signatures.
5) Figure 4 shows concentration values that do not correspond to the values in the text, on line 188.
6) Lines 233-237. Not quite an understandable sentence.
7) The hypothetical presence of clusters of gold nanoparticles on the cholesterol domains of the lipid membrane could be confirmed by microscopic studies. For example, by analyzing the surface using atomic force microscopy methods. Aggregates of nanoparticles would be clearly visible.
8) It is desirable to highlight as a detailed summarizing thesis exactly how, according to the authors, this method can be used for applied research.
Author Response
Dear Referee,
We would like to thank you for the review and suggestions. Please find enclosed the answers.

Reviewer 2 Report
The manuscript titled “Interaction between Nanoparticles, Membranes and Proteins: A Surface Plasmon Resonance Study” by de Macedo, E.F.; et al. is an original work where the authors address the adhesion and the dissociation rates of the complexes formed between two types of nanoparticles (NPs, gold and silver) with different composition of lipids that are the main precursors of membrane formation and proteins present in human blood serum. Surface Plasmon Resonance (SPR) is the main technique used to decipher the aforementioned properties. SPR technique has been widely employed previously to assess the interactions between a panoply of different matter systems. The scientific approach and methodology followed by the authors seem right and the gathered results can be relevant for the examined field. The most relevant outcomes of the present work are that both types of NPs poorly interact with 1-palmitoyl-2-oleoyl-glycero-3-phosphocholine (POPC) and the mix of POPC with dimethyldioctadecylammonium (POPC/DDAB) compared to POPC in presence of cholesterol. Moreover, gold and silver NPs showed greater affinity with fibrinogen and immunoglobulin G (IgG) proteins, respectively. The results achieved are well-discussed during the main body of the reported manuscript. The scientific paper is well written. In my opinion the present manuscript is innovative and the methodological approached used matches with the scope of International Journal of Molecular Sciences. For the above described reasons, I recommend the publication in International Journal of Molecular Sciences once the following remarks will be fixed:
--------
ABSTRACT
“(…) POPC only or with dimethyldioctadecylammonium (POC/DDAB)” (lines 25-26). Please, authors should modify the term “(POC/DDAB)” by “(POPC/DDAB)”.
“(…) IgG (…)” (line 29). Authors should introduce the definition of IgG abbreviation by adding “Immunoglobulin G”. Then, the term IgG should appear between brackets.
--------
KEYWORDS
(OPTIONAL). The keywords chosen by the authors are coherent respect the topic covered by the present manuscript. Nevertheless, I may suggest to add “biomolecular (or matter) interactions”, “blood serum proteins”, “complex dissociation rate” and/or “affinity”. Please, authors should take into account that the number of keywords upper limit is 10.
--------
INTRODUCTION
Introduction section is clear and concise. Furthermore, this section provides accurate information with relevant references of the system of study. Authors fully focused the Introduction section on the biomolecular interactions of NPs decorated with biomolecules of different nature and the host biomolecules (lines 43-112). Even if the article only studies the interaction of silver/gold nanoparticles with many biomolecules I miss a brief statement about the role of NPs size and geometry in cell toxicity. For example, it has been reported that gold NPs no larger than 6 nm effectively are uptaked by the cell nucleus, whereas larger NP diameter sizes (≥10 nm) only are found in the cytoplasm entering by the cell membrane [1]. This fact evidences that NPs several nanometers in size are more toxic than larger NPs which are unable to enter the nucleus.
[1] Huo, S.; et al. Ultrasmall gold nanoparticles as carriers for nucleus-based gene therapy due to size-dependent nuclear entry. ACS Nano 2014, 8, 5852-5862. https://doi.org/10.1021/nn5008572.
--------
RESULTS
The most significant outcomes are perfectly explained for all potential target audiences and other stakeholders. Nevertheless, the following points should be addressed:
I) “(…) (9.0 ± 0.4) mg/mm2 (…) (3.78 ± 0.06) and (3.8 ± 0.3) mg/mm2, respectively” (lines 187-188). Please, homogenize the significant figures. Same comment for “(10 ± 1) %; (9.2 ± 0.5) %; (7.3 ± 0.5) % and (5.6 ± 0.8) %” (lines 258-259).
II) “The enhanced interactions between gold NPs and the POPC/Chol membrane could be correlated with (…) as membrane model” (lines 189-193). Authors point out an interesting hypothesis to explain their results. Cholesterol is shown to disrupt POPC lipid layers in aqueous solution [2]. Could this fact partially be fine-in line with the results gathered by the authors? Authors should add a brief statement on this regard.
[2] Dai, J.; et al. Instability of cholesterol clusters in lipid bilayers and the cholesterol’s Umbrella effect. J. Phys. Chem B 2010, 114, 840-848. https://doi.org/10.1021/jp909061h.
III) Figure 6 (line 238). Maybe, it is an issue from the manuscript formatting but Figure 6 is slightly blurry. Authors should check what is the source of this issue and correct it in case to be necessary.
IV) Table 1 (line 279) and “Since the koff refers to the dissociation, the lower its value, the stronger the affinity between NPs and proteins.” (lines 281-282). I’m agree with the authors affirmation and I consider interesting the comparison they made with previous bibliography related to blood serum proteins (lines 307-313). Nevertheless, it may be relevant to introduce some biomolecular complexes found in nature with also high koff values. In this context, the koff values found for transferrin receptor:T7 peptide [3] and flavoenzyme:NADP+ cofactor [4] biomolecular systems being 6.8 10-4 and 2.0 10-2 s-1, respectively are also very low indicating the strong binding affinity of these complexes. Both complexes are well-known systems that play many functions in their respective living-organisms.
[3] Li, S.; et al. Evaluating the single-molecule interactions between targeted peptides and the receptors on living cell membrane. Nanoscale 2021, 13, 17318-17324. https://doi.org/10.1039/d1nr05547j.
[4] Pérez-Domínguez, S.; et al. Nanomechanical Study of Enzyme: Coenzyme Complexes: Bipartite Sites in Plastidic Ferredoxin-NADP+ Reductase for the Interaction with NADP. Antioxidants 2022, 11, 537. https://doi.org/10.3390/antiox11030537.
--------
DISCUSSION
Discussion is well structured and the most relevant results are highlighted. Authors should remark some potential futures perspectives in the use of SPR imaging for other biological systems and also conditional limitations in some specific cases. In the framework of future perspectives, authors should state the possibility to combine SPR imaging with other complementary techniques as bioluminescence resonance energy transfer (BRET) [5] or fluorescence cross-correlation spectroscopy (FCCS) [6] to assess the biomolecular dynamics and molecular recognition imaging (MRI) [7] to quantitatively map the specific biomolecular interactions, respectively.
[5] Verweij, E.W.E.; et al. BRET-Based Biosensors to Measure Agonist Efficacies in Histamine H1 Receptor-Mediated G Protein Activation, Signaling and Interactions with GRKs and β-Arrestins. Int. J. Mol. Sci. 2022, 23, 3184. https://doi.org/10.3390/ijms23063184.
[6] Jakobowska, I.; et al. Fluorescence Cross-Correlation Spectroscopy Yields True Affinity and Binding Kinetics of Plasmodium Lactate Transport Inhibitors. Pharmaceuticals 2021, 14, 757. https://doi.org/10.3390/ph14080757.
[7] Marcuello, C.; et al. Molecular Recognition of Proteins through Quantitative Force Maps at Single Molecule Level. Biomolecules 2022, 12, 594. https://doi.org/10.3390/biom12040594.
--------
MATERIALS AND METHODS
Authors have fully explained all the required information to carry out the same type of experiments in other labs from any world-wide research institution. Nevertheless, authors should indicate the country of the supplier for each purchased chemical compound or consumable materials. For example, authors perfect state “(Diener Electronic, Germany)” (line 563) or “(sciFLEXARRAYER, Germany)” (line 570), but it lacks this type of information on “(SPRi-BiochipsTM; Horiba)” (line 537) or “(Sigma-Aldrich)” line 539, among others.
--------
REFERENCES
Bibliography citations are not in the proper format of International Journal of Molecular Sciences. Authors should take care of this aspect.
--------
OVERVIEW AND FINAL COMMENTS
The submitted work is well-designed and the gathered results are interesting for the design and fabrication of next-generation of therapies based on drug delivery by NPs. For this reason, I will recommend the present scientific manuscript for further publication in International Journal of Molecular Sciences once all the aforementioned suggestions will be properly fixed.
Author Response

(The authors gave the same response as above.)
